# Mathematical Models to Describe the Foam Mat Drying Process of Cumbeba Pulp (*Tacinga inamoena*) and Product Quality

**DOI:** 10.3390/foods11121751

**Published:** 2022-06-14

**Authors:** Adelino de Melo Guimarães Diógenes, Rossana Maria Feitosa de Figueirêdo, Alexandre José de Melo Queiroz, João Paulo de Lima Ferreira, Wilton Pereira da Silva, Josivanda Palmeira Gomes, Francislaine Suelia dos Santos, Deise Souza de Castro, Marcela Nobre de Oliveira, Dyego da Costa Santos, Romário Oliveira de Andrade, Ana Raquel Carmo de Lima

**Affiliations:** 1Department of Technology in Agroindustry, Federal Institute of Education, Science, and Technology of Pernambuco, Afogados da Ingazeira 56800-000, Brazil; adelinoguimaraes@hotmail.com; 2Department of Agricultural Engineering, Federal University of Campina Grande, Campina Grande 58429-900, Brazil; alexandrejmq@gmail.com (A.J.d.M.Q.); joaop_l@hotmail.com (J.P.d.L.F.); wiltonps@uol.com.br (W.P.d.S.); josivanda@gmail.com (J.P.G.); francislainesuelis@gmail.com (F.S.d.S.); deise_castro01@hotmail.com (D.S.d.C.); 3Laboratory of Food Analysis, National Semiarid Institute, Campina Grande 58434-700, Brazil; marcela_nobre@msn.com; 4Department of Technology in Agroindustry, Federal Institute of Education, Science, and Technology of Rio Grande do Norte, Paus dos Ferros 59900-000, Brazil; dyego.csantos@gmail.com; 5Department of Technology in Agroindustry, Federal Institute of Education, Science, and Technology of de Alagoas, Piranhas 57460-000, Brazil; romario.andrade@ifal.com.br; 6Department of Technology in Agroindustry, Federal Institute of Education, Science, and Technology of de Alagoas, Batalha 57420-000, Brazil; ana.carmo@ifal.edu.br

**Keywords:** foam mat drying, mathematical modeling, diffusion coefficient, powder, color parameters

## Abstract

The present study investigated the mathematical modeling foam-mat drying kinetics of cumbeba pulp and the effect of drying conditions on the color and contents of ascorbic acid, flavonoids, and phenolic compounds of the powder pulps obtained. Foam-mat drying was carried out in a forced air circulation oven at temperatures of 50, 60, and 70 °C, testing foam-mat thicknesses of 0.5, 1.0, and 1.5 cm. The increase in the water removal rate is a result of the increase in air temperature and the decrease in the thickness of the foam layer. Among the empirical and semi-empirical mathematical models, the Midilli model was the one that best represented the drying curves in all conditions evaluated. Effective water diffusivity ranged from 1.037 × 10^−9^ to 6.103 × 10^−9^ m^2^ s^−1^, with activation energy of 25.212, 33.397, and 36.609 kJ mol^−1^ for foam thicknesses of 0.5, 1.0, and 1.5 cm, respectively. Cumbeba powders showed light orangish colors and, as the drying temperature increased from 50 to 70 °C, for all thicknesses, the lightness value (L*) decreased and the values of redness (+a*) and yellowness (+b*) increased. Foam-mat drying at higher temperatures (60 and 70 °C) improved the retention of ascorbic acid and flavonoids, but reduced the content of phenolic compounds, while the increase in thickness, especially for flavonoids and phenolic compounds, caused reduction in their contents. The foam-mat drying method allowed obtaining a good-quality cumbeba pulp powder.

## 1. Introduction

Cumbebeira (*Tacinga inamoena*) is a plant that belongs to the rustic cactus family and is well adapted to the climate and soil characteristics of the semi-arid region of Brazil. Its fruits, called cumbeba or quipá, present bioactive substances, including ascorbic acid, carotenoids, betalains, and phenolic and mineral compounds, with a relatively low acidity level [1,2,3,4,5,6,7]. However, cumbeba has a high water content, which is responsible for accelerating its deterioration. Thus, as an important factor in post-harvest management, it is necessary to employ technologies that allow its use for a longer period of time.

Hot air drying is one of the most used methods in the postharvest technology of agricultural products, aimed at extending their useful life, originating products with new characteristics and quality attributes. The most important alterations include the modification of sensory properties related to texture, flavor, aroma, and color, as well as of the nutritional content [8,9,10,11].

In the last years, different drying methods have been used to minimize changes in food caused by conventional hot air drying and maximize water removal, such as drying with microwaves, freeze drying, osmotic dehydration, drying by atomization, and foam-mat drying, among many others. In foam layer drying, a liquid or pasty material is transformed into a stable foam, obtained by the addition of foaming agents, and incorporation of nitrogen or air, usually followed by drying with hot air [12,13,14].

Due to its simplicity, low cost, and high rate of water removal, combined to the good quality of the products obtained, foam-mat drying has been successfully used in the production of powders of juice and/or pulp of fruits such as pineapple [15], cherry [16], mango [17], jambolan (*Syzygium cumini* L.) [18], and melon [19]. It has also been applied to obtain powders from mushroom (*Agaricus bisporus*) puree [20], yogurt [21], shrimp [22], yacon juice (*Smallanthus sonchifolius*) [23], and beet (*Beta vulgaris*) [24].

The present study evaluated the foam-mat drying of cumbeba pulp at different temperatures and foam layer thicknesses, determining the drying kinetics, effective moisture diffusivity, activation energy, and the quality attributes color, vitamin C, flavonoids, and phenolic compounds of the powders produced under the different drying conditions.

## 2. Materials and Methods

### 2.1. Materials

Cumbebeira fruits (*T. inamoena*) at a mature stage of maturation, identified by the color of the yellow-orange skin, were used as raw material. The fruits were harvested in the municipality of Afogados da Ingazeira, State of Pernambuco, Brazil. The average initial moisture was 8.43 g g^−1^, on a dry basis (d.b.), determined in a vacuum oven at 70 °C (AOAC) [25]. The additives used for foam formation were Emustab (emulsifier) and Liga Neutra (stabilizer) (Du Porto^®^, Porto Feliz, São Paulo, Brazil).

### 2.2. Fruit Processing

Cumbeba fruits were washed, brushed to remove thorns, and sanitized by immersion in chlorine solution (100 ppm) for 30 min. The pulp was extracted using a stainless-steel pulper (Laboremus, Campina Grande, Paraíba, Brazil).

### 2.3. Foam Preparation

The cumbeba pulp was mixed with the emulsifier (2.5 g 100 g^−1^ of pulp) and the stabilizer (1.5 g 100 g^−1^ of pulp) and stirred in a domestic mixer (Arno, model SX15, São Paulo, Brazil) at maximum speed (rotation speed level: 3) for 15 min for foam formation.

### 2.4. Foam-Mat Drying

The foam was distributed in stainless steel trays of 12 cm in diameter, forming layers with thicknesses of 0.5, 1.0, and 1.5 cm, and dried in triplicate in an oven with forced air circulation (Fanem, model 320, Guarulhos, São Paulo, Brazil) at temperatures of 50, 60, and 70 °C, with relative humidity of 67, 62, and 60%, respectively, and air velocity of 1.0 m s^−1^. In the environment, the temperature and relative humidity were on average 27.33 °C and 76.67%, respectively. The trays with the foam were weighed on a semi-analytical scale (Marte, model AS5500C, Santa Rita do Sapucaí, Minas Gerais, Brazil) at intervals of 5, 10, 20, 30, and 60 min until reaching constant weight in three consecutive weighings, considered as a steady state. Subsequently, the samples were dried in a vacuum oven at 70 °C to obtain the dry mass, used to determine the water content at each time point [25]. After drying, the dry foam was removed from the trays and pulverized in a domestic processor (Black Decker, Model HC31X-Type 2, Uberaba, Minas Gerais, Brazil), until powder was obtained.

Data of the drying kinetics of the foams were used to calculate the moisture content ratios (Equation (1)) [26,27].
(1)$$\mathrm{MR}=\frac{M_{t}-M_{e}}{M_{i}-M_{e}}$$
where MR is the moisture content ratio (dimensionless), M_t_, M_i_, and M_e_ are the moisture content at time t, initial moisture content and equilibrium moisture content (g 100 g^−1^), respectively, on a dry basis.

The drying rate −dM/dt for each experiment can be obtained through the derivative of Equation (1) with respect to time [28]:(2)$$-\frac{\mathrm{dM}}{\mathrm{dt}}=\left(M_{i}-M_{e}\right)\left[-\frac{\mathrm{dMR}}{\mathrm{dt}}\right]$$

### 2.5. Mathematical Modelling of the Foam-Mat Drying Process

Mathematical models (Table 1) were fitted to the experimental data of foam drying kinetics, using the computer program Statistica version 7.0 (StatSoft^®^ Inc., Tulsa, OK, USA).

To assess the quality of the models fit to the experimental data, the coefficient of determination (R^2^) (Equation (3)), the mean square deviation (MSD) (Equation (4)) and the chi-square (χ^2^) (Equation (5)) were determined [38,39]. After selecting the best model, it was used to determine the drying rates at each temperature.
(3)$$R^{2}=\frac{\sum_{i=1}^{N}{\left[\left({\mathrm{MR}}_{\exp,i}-{\bar{\mathrm{MR}}}_{\exp,i}\right)\left({\mathrm{MR}}_{\mathrm{pred},i}-{\bar{\mathrm{MR}}}_{\mathrm{pred},i}\right)\right]}^{2}}{\sum_{i=1}^{N}{\left({\mathrm{MR}}_{\exp,i}-{\bar{\mathrm{MR}}}_{\exp,i}\right)}^{2}\sum_{i=1}^{N}{\left({\mathrm{MR}}_{\mathrm{pred},i}-{\bar{\mathrm{MR}}}_{\mathrm{pred},i}\right)}^{2}}$$
(4)$$\mathrm{MSD}={\left[\frac{1}{N}\overset{N}{\underset{i=1}{\sum }}{\left({\mathrm{MR}}_{\mathrm{pred},i}-{\mathrm{MR}}_{\exp,i}\right)}^{2}\right]}^{\frac{1}{2}}$$
(5)$$\chi^{2}=\frac{1}{N-n}\overset{N}{\underset{i=1}{\sum }}{\left({\mathrm{MR}}_{\mathrm{pred},i}-{\mathrm{MR}}_{\exp,i}\right)}^{2}$$
where MR_exp,i_ is the experimental moisture content ratio, ${\bar{\mathrm{MR}}}_{\exp,i}$ is the mean of the experimental moisture content ratio, MR_pred,i_ is the moisture content ratio predicted by the model, ${\bar{\mathrm{MR}}}_{\mathrm{pred},i}$ is the mean of the moisture content ratio predicted by the model, N is the number of experimental points, and n is the number of constants of the model.

### 2.6. Determination of Effective Moisture Diffusivity and Activation Energy

Effective moisture diffusivity was determined by fitting the mathematical model of liquid diffusion with eight-term approximation (Equation (6)) to the experimental data of foam drying kinetics at different temperatures, considering uniform initial moisture distribution, constant diffusivity, and negligible thermal resistance and volumetric shrinkage. This model is the analytical solution for Fick’s second law considering the geometric shape of the foam layers approximately as a flat plate (area >> thickness) [40].
(6)$$\mathrm{MR}=\frac{M_{t}-M_{e}}{M_{i}-M_{e}}=\frac{8}{\pi^{2}}\overset{\infty }{\underset{n=0}{\sum }}\frac{1}{{\left(2n+1\right)}^{2}}\exp\left[-{\left(2n+1\right)}^{2}\pi^{2}D_{\mathrm{eff}}\frac{t}{{4L}^{2}}\right]$$
where D_eff_ is the effective diffusion coefficient (m^2^ s^−1^), n is the number of terms in the equation, L is the half thickness of the foam (m), and t is the time (s).

The relationship between effective moisture diffusivity and drying temperatures was described by an Arrhenius-type equation (Equation (7)) [41,42,43].
(7)$$D_{\mathrm{eff}}{=D}_{0}\left(-\frac{E_{e}}{R\left(T+273.15\right)}\right)$$
where D_0_ is the pre-exponential factor (m^2^ s^−1^), E_a_ is the activation energy (J mol^−1^), R is the universal gas constant (8.314 J mol^−1^ K^−1^), and T is the temperature (°C).

Arrhenius-type equation parameters were obtained by linearizing Equation (7), applying the natural logarithm, according to Equation (8) [44,45].
(8)$${\mathrm{LnD}}_{\mathrm{eff}}={\mathrm{LnD}}_{0}\left(-\frac{E_{a}}{R\left(T+273.15\right)}\right)$$
where LnD_0_ is the logarithmic of the pre-exponential factor (m^2^ s^−1^), E_a_ is the activation energy (J mol^−1^), R is the universal gas constant (8.314 J mol^−1^ K^−1^), and T is the temperature (°C).

### 2.7. Physicochemical Properties of the Powders

#### 2.7.1. Color Measurement

The color parameters (L*, a*, b*) of the cumbeba pulp foam powders were measured using a portable spectrophotometer (Hunterlab, XE Plus model, Reston, VA, USA). L* indicates lightness (0 = black and 100 = white), while a* and b* indicate chromaticity (−a*: greenness and +a*: redness; −b*: blueness and +b*: yellowness). The device was calibrated with standard white and black plates. Readings in the samples were taken in in quadruplicate using the system with illuminant D65.

#### 2.7.2. Vitamin C (VC)

The vitamin C content was determined using the titration method, based on the reduction of 2,6-dichlorophenolindophenol in the presence of oxalic acid, according to Benassi and Antunes [46]. The result was expressed as mg of vitamin C 100 g^−1^ of sample in dry matter. The dichlorophenolindophenol solution was standardized with vitamin C solution (0.5 g L^−1^). The analysis was performed in quadruplicate.

#### 2.7.3. Total Flavonoids (TF)

The total flavonoid (TF) content was determined, in quadruplicate, according to the Francis methodology [47]. The extract was obtained by macerating the sample (0.065 g) with 10 mL of ethanol-HCl solution (1.5 N) in the proportion of 85:15 (*v*/*v*) for 1 min, followed by resting at 5 °C/24 h. The extract was filtered through cotton and the absorbance was read in a UV-Vis spectrophotometer (Coleman, Model 35-D, Santo André, São Paulo, Brazil) at 374 nm. The result was expressed in mg 100 g^−1^ of sample in dry matter.

#### 2.7.4. Total Phenolic Compounds (TPC)

The content of total phenolic compounds (TPC) was determined, in quadruplicate, according to the Folin–Ciocalteu method [48]. Initially 1.0 g of the sample was homogenized with 50 mL of distilled water; then, it was left to rest for 30 min in the absence of light and at room temperature (25 ± 2 °C). Afterwards, the extract was filtered and the absorbance was read in a UV-Vis spectrophotometer (Coleman, Model 35-D, Santo André, São Paulo, Brazil) at 765 nm. The result was expressed in mg of gallic acid equivalent per 100 g of dry matter sample, with the standard curve obtained with the gallic acid solution (0–22.5 µg mL^−1^).

### 2.8. Statistical Analysis

The experimental data were verified statistically by analysis of variance (ANOVA) with Tukey test was applied to analyze the differences between treatment means at 95% confidence level. The calculations were performed using computer program Statistica version 7.0 (StatSoft^®^ Inc., Tulsa, OK, USA).

## 3. Results and Discussion

### 3.1. Cumbeba Pulp Foam-Mat Drying Kinetics

The experimental curves of cumbeba pulp foam-mat drying kinetics for layer thicknesses of 0.5, 1.0, and 1.5 cm, at temperatures of 50, 60, and 70 °C, which describe the evolution of moisture content ratio as a function of time, are shown in Figure 1. There were reductions with time, with increasing drying temperature (Figure 1a–c) and with decreasing layer thickness (Figure 1d–f). The drying times corresponded to 460, 310, and 280 min, for the 0.5 cm thickness and temperatures of 50, 60, and 70 °C, with final moisture contents of 13.18, 8.19, and 7.60% (d.b.), respectively. For the 1.0 cm thickness, at these three temperatures, the drying times were 880, 760, and 580 min, with final moisture contents of 13.76, 8.81, and 8.50% (d.b.), while for the 1.5 cm thickness the drying times were 1300, 880, and 820 min with final moisture contents of 14.19, 12.16, and 10.39% (d.b.), at temperatures of 50, 60, and 70 °C, respectively.

The increment of temperature results in the increase in heat transfer rate [49], which leads to an increased agitation of water molecules [14], which translates into an increment of their mobility [50]. In addition, the reduction of thickness decreases the distance that the water has to travel from the interior to the surface of the foam [51], reducing the drying time. Similar results were reported for the foam-mat drying of banana puree [52], papaya pulp foam [53], melon pulp foam [54], tamarind juice foam [55], and atemoya pulp foam [56].

The drying rates of the foams, under the different drying conditions, are shown in Figure 2. For all temperatures and thicknesses studied, the drying rates were higher at the beginning of the process, which shows a short period of fast drying that is evidenced by an initial increasing drying rate until a maximum value. On the other hand, the drying of the samples occurred mainly in the period of decreasing rate with time, not being observed a period of constant rate. When the surface of the product dries quickly and the diffusion of water molecules becomes less than its external convection, the drying front migrates towards the interior of the product [57], and the internal resistance to the movement of molecules of water becomes the dominant transport mechanism, as a result the drying rate decreases [58,59]. Similar results were found in the drying of mango pulp foam [60], papaya pulp foam [61], pumpkin pulp foam [62], and tomato pulp foam [63].

### 3.2. Mathematical Modeling of the Drying Kinetics of the Foams

Table 2 presents the coefficients of the models fitted to the data of drying kinetics of the foams and the values of the parameters (R^2^, MSD and χ^2^) used to determine the quality of fit. All models used had R^2^ values higher than 0.980, with MSD and χ^2^ values lower than 0.0448 and 0.0021, respectively. Among all models tested, the Midilli model showed the highest values of R^2^ (0.9972–0.9990) and lowest values of MSD (0.0112–0.0185) and χ^2^ (0.0002–0.0004), so this model represents better the drying process of the foams. Good fits obtained with the Midilli model are frequently reported in the literature, such as in the drying of beet pulp [64], soursop pulp foam [65], and guava pulp foam [66].

Based on the comparison presented between the experimental data of the water content ratio and the data predicted by the Midilli model (Figure 3), it appears that the data predicted by the Midilli model are distributed around the straight line that represents the equality between the MR predicted by the model and the experimental MR, showing the adequacy of the model in the description of the loss of water during the drying process of the cumbeba pulp foam in all the conditions studied. Empirical and semi-empirical mathematical models are suggested as suitable to represent the drying process of a product when the external resistance to heat and mass transfer is eliminated or negligible [37].

For Midilli model, the derivative −dMR/dt can be determined by the following expression:(9)$$-\frac{\mathrm{dMR}}{\mathrm{dt}}=a\;\exp\left(-{kt}^{n}\right)\left(-{\mathrm{knt}}^{n-1}\right)-b$$
where the parameters a,b, k, and n were given in Table 2. Thus, the graph of Figure 2 showing −dMR/dt versus t can be obtained for all experiments. As additional information, the graph shown in Figure 2 was provided by LAB Fit Curve Fitting Software version 7.2.50b (Federal University of Campina Grande, Campina Grande, Paraíba, Brazil) [67].

### 3.3. Effective Moisture Diffusivity and Activation Energy

Naturally, during the drying process, especially in the initial moments, the temperature of the product and the drying air are different, but when the water removal process progresses, the temperature of the product gradually increases until a state of thermal equilibrium is reached. In fact, in this study we focused our attention only on the mass (water) transfer mechanisms; thus, we consider that in each experiment the drying process occurs isothermally. The change in the effective moisture diffusivity (D_eff_) with temperature, showing the fit of the Arrhenius-type equation (Equation (7)), for each foam layer thickness is shown in Figure 4. D_eff_ values ranging from 1.037 × 10^−9^ m^2^ s^−1^ (0.5 cm/50 °C) to 6.103 × 10^−9^ m^2^ s^−1^ (1.5 cm/70 °C), evidencing the influence of the drying conditions. The trend of increase in D_eff_ with the increments of temperature and foam layer thickness is explained by the increase in the heat transfer rate between the foam and the drying air, due to the increase in temperature, which results in greater agitation of water molecules [16,68] and its diffusion. As thickness increases, there is a reduction of entropy [19] and formation of a foam with more organized and uniform internal structure [22,56,69], which results in a more efficient system in the molecular transport of water, hence the higher diffusivity, as observed in the works of Kadam and Balasubramanian [70] in the drying of tomato juice foam, Wilson et al. [71], in the drying of mango pulp foam, Sousa et al. [72] in the drying of pequi pulp, and Dehghannya et al. [73] in the drying of lemon juice foam.

The results of fitting Equation (7) in pairs (T, D_eef_) are shown in Table 3. The values of D_0_, E_a_/R and R^2^ were, respectively, 1.2880 × 10^−5^ m^2^ s^−1^, 3032.5043 K, and 0.9291 for the thickness of 0.5 cm, 4.0785 × 10^−4^ m^2^ s^−1^, 4017.0737 K, and 0.9840 for the thickness of 1.0 cm and 2.355 × 10^−3^ m^2^ s^−1^, 4403.3859 K, and 0.9855 for the thickness of 1.5 cm. The values of activation energy (E_a_) of the foams (Table 3) were calculated from the slope of the curve of the Arrhenius-type equation (Equation (8)). When associated with the drying process, the activation energy can be understood as the energy that must be supplied to the product for water diffusion to start [74]. The activation energy (E_a_) for liquid diffusion of the foams under the different drying conditions ranged from 25.21 to 36.61 kJ mol^−1^, for the thicknesses of 0.5 and 1.5 cm, respectively. These values are close to the E_a_ values reported by Thuwapanichayanan et al. [75] in the drying of banana pulp foam (21.08 to 25.19 kJ mol^−1^), by Gupta and Alam [76] in drying of concentrated grape juice (36.35 kJ mol^−1^), and by Salahi et al. [77], who reported E_a_ values of 31.714 and 33.043 kJ mol^−1^ in the drying of melon pulp foam, with thicknesses of 3 and 5 cm, respectively.

### 3.4. Physicochemical Properties of the Powders

#### 3.4.1. Color

The results concerning the effect of drying conditions on the color parameters of the powders obtained from the foams are presented in Table 4. The powders showed L* values ranging from 55.55 (70 °C/1.5 cm) to 63.0 (50 °C/0.5 cm), being significantly (*p* < 0.05) influenced by the drying conditions; increments of temperature and foam thickness caused a reduction in L* values. A possible explanation for this behavior is that prolonged heating in the presence of moisture favors reactions that result in the formation of dark compounds, such as the Maillard reaction and oxidation of vitamin C. Franco et al. [23] observed the same behavior during the drying of the yacon juice foam, where the L* values decreased with the increments of temperature and thickness of the samples.

In relation to the parameter a* (from green −a* to red +a*), the powders showed positive values, varying from 11.57 (50 °C/0.5 cm) to 13.51 (70 °C/1.5 cm), in the red region. The drying conditions significantly influenced (*p* < 0.05) the values of a* (Table 4), which increased with the increments in air temperature and foam thickness. The drying conditions also significantly affected (*p* < 0.05) the parameter b* (transition from blue −b* to yellow +b*), which tended to increase with the increments in temperature and foam thickness, with values varying from 44.66 (50 °C/0.5 cm) to 49.26 (70 °C/1.5 cm), results that characterize the predominance of yellow hue in all powders. Increments in a* and b* values may be associated with the thermal inactivation of enzymes that interfere in the oxidation of natural pigments (carotenoids and betalains) [4,5] present in the cumbeba pulp and/or in the concentration of these pigments in the powders, as well as with the occurrence of Maillard reactions [22]. Salahi et al. [77] reported similar behavior in the drying of melon foam. These authors observed that the values of the color parameters (+a* and +b*) increased as a function the increase in drying temperature and foam thickness.

#### 3.4.2. Vitamin C (VC)

The VC content ranged from 47.80 mg 100 g^−1^ (50 °C/1.5 cm) to 92.38 mg 100 g^−1^ (70 °C/1.5 cm) (Table 4). The results showed that the VC content of the powders was affected by the drying conditions, but was significantly affected (*p* < 0.05) only by the drying temperature, which caused a tendency of retention of the vitamin C content. This behavior can be attributed to the reduction of process time as the temperature increased, which may have led to the reduction in VC degradation rate. Kandasamy et al. [78] emphasize that prolonged heat treatment may cause oxidation of VC. Santos et al. [79], evaluating the influence of drying conditions on the quality properties of white pitaya peel powders, also observed the concentration of VC content with the elevation of the drying temperature.

#### 3.4.3. Total Flavonoids (TF)

The TF content ranged from 101.53 mg 100 g^−1^ (50 °C/1.5 cm) to 161.51 mg 100 g^−1^ (70 °C/1.5 cm) (Table 4). The concentration of TF in the powders was significantly (*p* < 0.05) influenced by the drying conditions. At a temperature of 50 °C, it tended to decrease with the increase in thickness and, at temperatures of 60 and 70 °C, it increased, suggesting that the drying temperature positively affected the flavonoids content, possibly due to the reduction of drying time. A possible explanation for this result is that the reduction of drying time by the use of higher temperatures may have minimized the deleterious effects of the degradation reactions [80]. Moussa-Ayoub et al. [81], studying extrusion of cactus pear pulp at temperatures of 100, 140, and 160 °C, also reported increase in the TF content with the elevation of temperature.

#### 3.4.4. Total Phenolic Compounds (TPC)

The TPC content ranged from 552.59 mg 100 g^−1^ (50 °C/1.5 cm) to 1334.48 mg 100 g^−1^ (70 °C/0.5 cm) (Table 4). The results showed that the TPC content of the powders was significantly (*p* < 0.05) affected by the drying conditions. With the increment in drying temperature, the concentration of phenolic compounds increased, decreasing with layer thickness. Reduction of phenolic compounds may occur because they are involved in protein complexation reactions [82] and/or enzymatic oxidation [83], both stimulated by the elevation of temperature and prolonged exposure to oxygen. Reduction in these compounds has been previously observed by Chandrasekar et al. [84], in the foam-mat drying of juice, prepared by mixing bitter gourd, cucumber, tomato juice, and water in the proportion of 30:30:30:10, respectively, and by Auisakchaiyoung and Rojanakorn [85], in the foam-mat drying of Gac fruit (*Momordica cochinchinensis*). The increase with temperature, as in the case of flavonoids, may be a consequence of the shorter time of exposure to heating.

## 4. Conclusions

The drying of the cumbeba pulp foam occurred mainly in the period of decreasing rate. The Midilli model was the one that best described the drying behavior in the different conditions evaluated. The effective diffusivity in the foams increased with increasing air temperature and foam thickness, with activation energy ranging from 25.212 to 36.609 kJ mol^−1^. Redness (+a*), yellowness (+b*), Vitamin C, and flavonoids of the cumbeba pulp foam powders were sensitive to the drying temperature; increment of temperature causes increase in +a*, +b* and contents of vitamin C, flavonoids, and phenolic compounds, and reduction of L*. The increase in thickness caused a reduction in phenolic compounds at all temperatures and an increase in the contents of vitamin C and flavonoids at higher temperatures. The foam-mat drying proved to be a viable method for processing cumbeba pulp, since it allowed to obtain a product (powder) that can be used as food, either through pure consumption or in the preparation of juices, gelatin, yogurt, ice cream, natural coloring, and as an ingredient in confectionery and bakery products.

## Figures and Tables

**Figure 1 foods-11-01751-f001:**
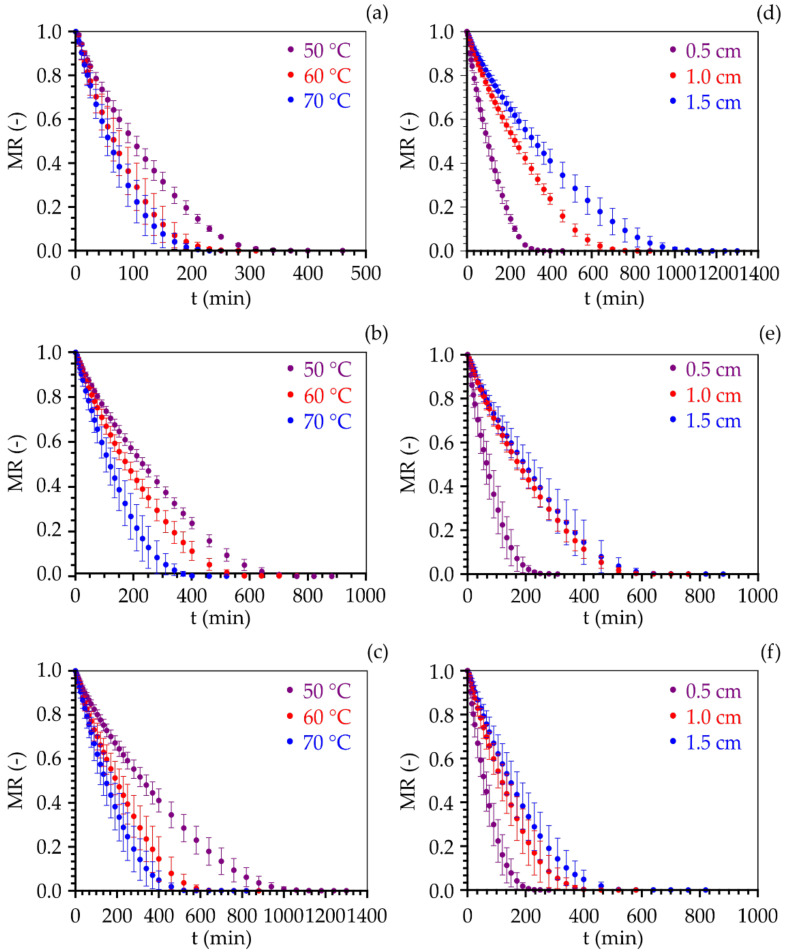
Temporal evolution of moisture content ratio in the foam samples with different thicknesses ((**a**) 0.5 cm, (**b**) 1.0 cm, and (**c**) 1.5 cm) and air temperatures ((**d**) 50 °C, (**e**) 60 °C, and (**f**) 70 °C). Vertical bars represent the standard deviation of n = 3 repetitions.

**Figure 2 foods-11-01751-f002:**
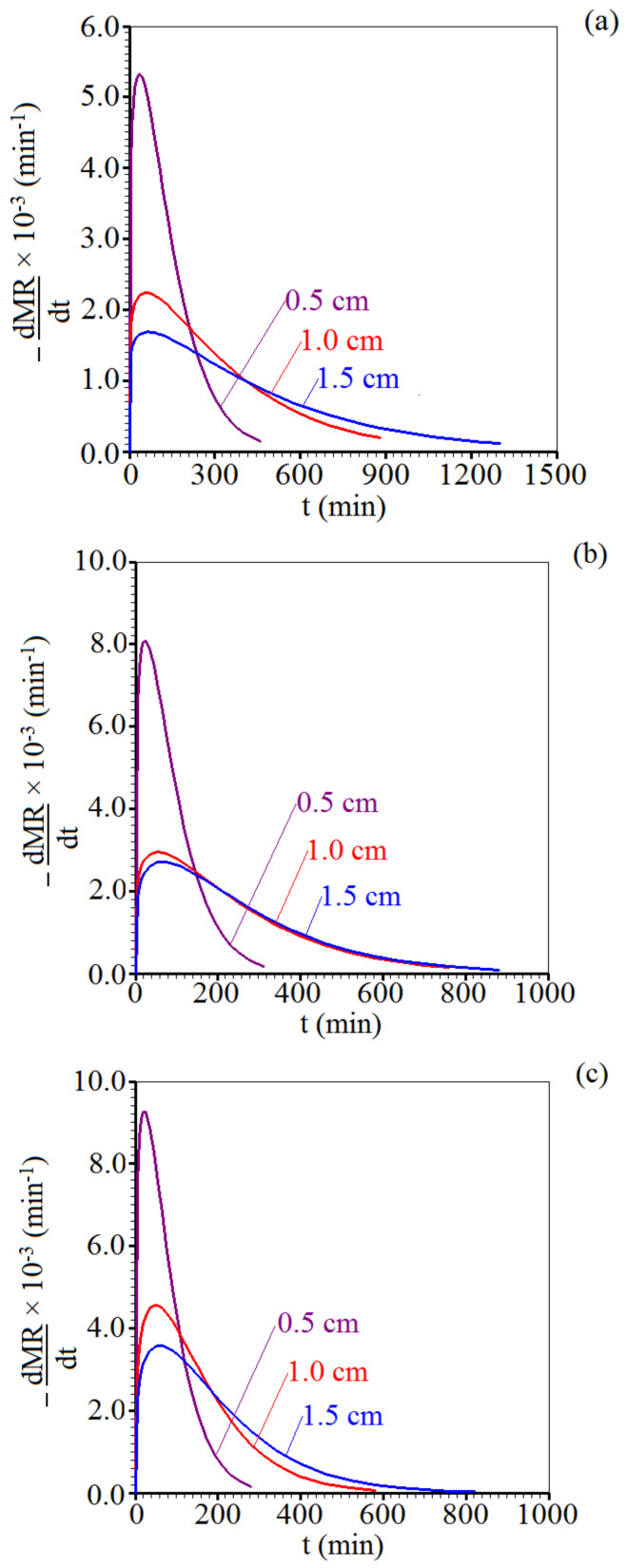
Effect of air temperature and sample thickness on the evolution of the drying rate of the foams correspond to (**a**) 50, (**b**) 60, and (**c**) 70 °C, respectively.

**Figure 3 foods-11-01751-f003:**
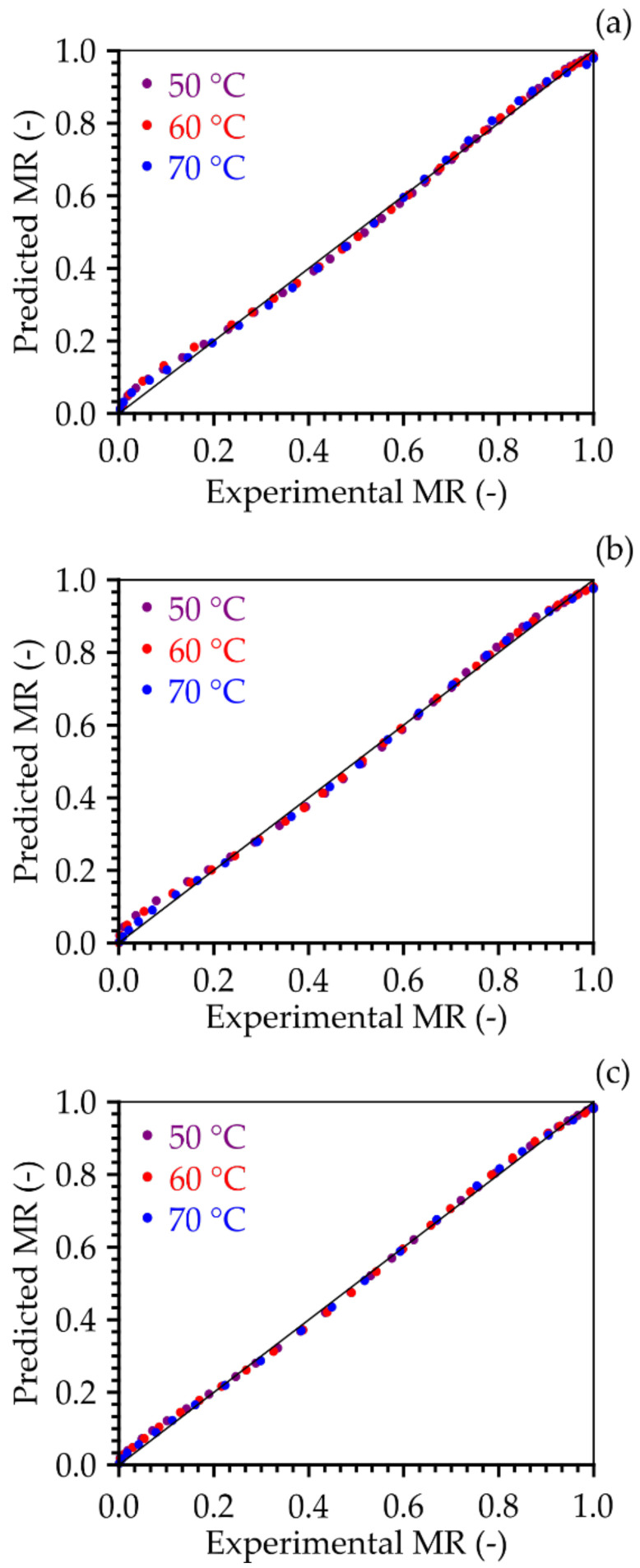
Moisture content ratio predicted by the Midilli model versus experimental moisture content ratio for (**a**) 0.5 cm, (**b**) 1.0 cm, and (**c**) 1.5 cm. The solid curve represents the regression line (MR _Predicted_ = MR _experimental_).

**Figure 4 foods-11-01751-f004:**
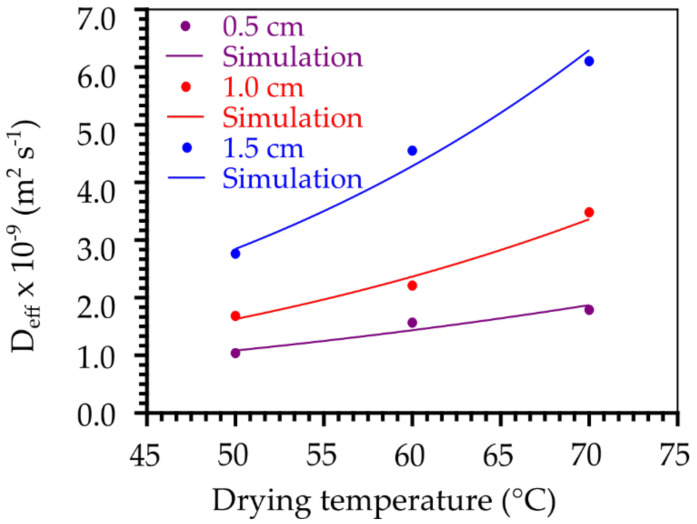
Effective moisture diffusivity obtained under different drying conditions of cumbeba pulp foam.

**Table 1 foods-11-01751-t001:** Mathematical models used to estimate the drying kinetics curves of the foams.

Model n^o^.	Model Name	Model Equation	References
1	Newton	MR=exp(−kt)	[29]
2	Page	$\mathrm{MR}=\exp(-{\mathrm{kt}}^{n})$	[30]
3	Henderson and Pabis	$\mathrm{MR}=a\;\exp\left(-\mathrm{kt}\right)$	[31]
4	Exponential of two terms	$\mathrm{MR}=a\;\exp(-\mathrm{kt})+\left(1-a\right)\exp(-\mathrm{kat})$	[32]
5	Thompson	$\mathrm{MR}=\exp(-a\;(a^{2}+4\mathrm{bt}{)}^{0.5})/2b)$	[33]
6	Logarithmic	MR=a exp(−kt)+c	[34]
7	Diffusion approach	$\mathrm{MR}=a\;\exp(-\mathrm{kt})+\left(1-a\right)\exp(-\mathrm{kbt})$	[32]
8	Henderson and Pabis modified	$\mathrm{MR}=a\;\exp(-\mathrm{kt})+b\;\exp(-\mathrm{kt})+c\;\exp\left(-\mathrm{kt}\right)$	[35]
9	Two terms	$\mathrm{MR}=a\;\exp(-k_{0}t)+b\;\exp(-t)$	[36]
10	Midilli	$\mathrm{MR}=a\;\exp\left(-{\mathrm{kt}\;}^{n}\right)+\mathrm{bt}$	[37]

MR—moisture content ratio, dimensionless; k—drying constants; a, b, c, k_0_, k_1_, n—coefficients of the models; t—drying time (min).

**Table 2 foods-11-01751-t002:** Coefficients of determination (R^2^), mean squared deviations (MSD) and chi-square (χ^2^) for the models fitted to the data of foam-mat drying kinetics.

Model	Temp. (°C)	Foam Thickness (cm)	Parameters of Model	R^2^	MSD	χ^2^
Newton	50	0.5	k: 0.0079	0.9853	0.0428	0.0019
		1.0	k: 0.0033	0.9830	0.0446	0.0021
		1.5	k: 0.0024	0.9859	0.0419	0.0002
	60	0.5	k: 0.0118	0.9874	0.0393	0.0016
		1.0	k: 0.0043	0.9870	0.0391	0.0016
		1.5	k: 0.0039	0.9854	0.0425	0.0019
	70	0.5	k: 0.0135	0.9882	0.0383	0.0017
		1.0	k: 0.0066	0.9845	0.0444	0.0020
		1.5	k: 0.0052	0.9874	0.0406	0.0017
Page	50	0.5	k: 0.0025; n: 1.2350	0.9957	0.0232	0.0006
		1.0	k: 0.0010; n: 1.2109	0.9939	0.0268	0.0008
		1.5	k: 0.0006; n: 1.2131	0.9948	0.0259	0.0007
	60	0.5	k: 0.0043; n: 1.2240	0.9970	0.0190	0.0004
		1.0	k: 0.0014; n: 1.2056	0.9955	0.0228	0.0006
		1.5	k: 0.0011; n: 1.2293	0.9950	0.0247	0.0006
	70	0.5	k: 0.0048; n: 1.2375	0.9982	0.0148	0.0003
		1.0	k: 0.0017; n: 1.2737	0.9974	0.0181	0.0004
		1.5	k: 0.0013; n: 1.2591	0.9982	0.0153	0.0002
Henderson and Pabis	50	0.5	a: 1.0443; k: 0.0083	0.9882	0.0383	0.0016
		1.0	a: 1.0318; k: 0.0034	0.9851	0.0418	0.0019
		1.5	a: 1.0168; k: 0.0072	0.9873	0.0403	0.0017
	60	0.5	a: 1.0439; k: 0.0125	0.9900	0.0349	0.0013
		1.0	a: 1.0317; k: 0.0045	0.9889	0.0361	0.0014
		1.5	a: 1.0343; k: 0.0041	0.9875	0.0393	0.0016
	70	0.5	a: 1.0519; k: 0.0143	0.9915	0.0325	0.0014
		1.0	a: 1.0518; k: 0.0071	0.9885	0.0382	0.0016
		1.5	a: 1.0487; k: 0.0056	0.9909	0.0345	0.0013
Exponential of two terms	50	0.5	a: 1.7579; k: 0.0111	0.9954	0.0239	0.0006
		1.0	a: 1.7535; k: 0.0047	0.9939	0.0268	0.0008
		1.5	a: 1.7340; k: 0.0033	0.9949	0.0256	0.0007
	60	0.5	a: 1.7506; k: 0.0165	0.9969	0.0195	0.0004
		1.0	a: 1.7287; k: 0.0060	0.9956	0.0226	0.0005
		1.5	a: 1.7503; k: 0.0056	0.9950	0.0249	0.0007
	70	0.5	a: 1.7708; k: 0.0190	0.9979	0.0160	0.0003
		1.0	a: 1.8023; k: 0.0096	0.9969	0.0198	0.0004
		1.5	a: 1.7929; k: 0.0076	0.9979	0.0165	0.0003
Thompsom	50	0.5	a: −464.982; b: 1.9214	0.9851	0.0430	0.0020
		1.0	a: −588.053; b: 1.4036	0.9829	0.0448	0.0021
		1.5	a: −712.785; b: 1.3149	0.9858	0.0426	0.0019
	60	0.5	a: −377.329; b: 2.1202	0.9872	0.0395	0.0017
		1.0	a: −769.019; b: 1.8307	0.9869	0.0392	0.0016
		1.5	a: −532.336; b: 1.4582	0.9852	0.0427	0.0019
	70	0.5	a: −438.664; b: 2.4363	0.9881	0.0386	0.0020
		1.0	a: −508.795; b: 1.8468	0.9844	0.0446	0.0021
		1.5	a: −593.769; b: 1.7694	0.9873	0.0407	0.0018
Logarithm	50	0.5	a: 1.1302; k: 0.0066; c: −0.1089	0.9958	0.0226	0.0006
		1.0	a: 1.1953; k: 0.0024; c: −0.1894	0.9966	0.0198	0.0004
		1.5	a: 1.1595; k: 0.0018; c: −0.1575	0.9972	0.0189	0.0004
	60	0.5	a: 1.1125; k: 0.0101; c: −0.0912	0.9966	0.0204	0.0005
		1.0	a: 1.1396; k: 0.0035; c: −0.1290	0.9968	0.0194	0.0004
		1.5	a: 1.1302; k: 0.0033; c: −0.1156	0.9951	0.0244	0.0007
	70	0.5	a: 1.0985; k: 0.0123; c: −0.0647	0.9962	0.0216	0.0007
		1.0	a: 1.1271; k: 0.0058; c: −0.0945	0.9950	0.0252	0.0007
		1.5	a: 1.1005; k: 0.0048; c: −0.0651	0.9950	0.0255	0.0007
Diffusion approach	50	0.5	a: −0.1288; k: 0.0573; b: 0.1561	0.9915	0.0325	0.0012
		1.0	a: −6.6210; k: 0.0062; b: 0.9127	0.9945	0.0252	0.0007
		1.5	a: −14.580; k: 0.0043; b: 0.9593	0.9954	0.0240	0.0006
	60	0.5	a: −12.652; k: 0.0061; b: 1.0513	0.9972	0.0183	0.0004
		1.0	a: −25.349; k: 0.0021; b: 1.0294	0.9975	0.0171	0.0003
		1.5	a: −71.321; k: 0.0020; b: 1.0101	0.9995	0.0079	0.0001
	70	0.5	a: −7.8441; k: 0.0248; b: 0.9233	0.9983	0.0142	0.0003
		1.0	a: −7.6219; k: 0.0128; b: 0.9166	0.9974	0.0179	0.0004
		1.5	a: −7.2622; k: 0.0101; b: 0.9144	0.9983	0.0148	0.0002
Henderson and Pabis modified	50	0.5	a: 0.3489; k: 0.0083; b: 0.3477; c: 0.3477	0.9882	0.0383	0.0017
		1.0	a: 0.3452; k: 0.0034; b: 0.3433; c: 0.3433	0.9851	0.0418	0.0020
		1.5	a: 0.3461; k: 0.0025; b: 0.3404; c: 0.3404	0.9873	0.0403	0.0018
	60	0.5	a: 0.3525; k: 0.0125; b: 0.3456; c: 0.3456	0.9900	0.0349	0.0015
		1.0	a: 0.3461; k: 0.0045; b: 0.3427; c: 0.3427	0.9889	0.0361	0.0015
		1.5	a: 0.3458; k: 0.0041; b: 0.3442; c: 0.3442	0.9875	0.0393	0.0018
	70	0.5	a: 0.3512; k: 0.0143; b: 0.3503; c: 0.3503	0.9915	0.0325	0.0016
		1.0	a: 0.3528; k: 0.0071; b: 0.3494; c: 0.3494	0.9885	0.0382	0.0017
		1.5	a: 0.2630; k: 0.0060; b: 0.3530; c: 0.3530	0.9909	0.0345	0.0014
Two terms	50	0.5	a: 0.6795; k_0_: 0.0083; b: 0.3648; k_1_: 0.0083	0.9882	0.0383	0.0017
		1.0	a: 0.3721; k_0_: 0.0034; b: 0.6597; k_1_: 0.0034	0.9851	0.0418	0.0020
		1.5	a: 0.3469; k_0_: 0.0025; b: 0.6800; k_1_: 0.0025	0.9873	0.0403	0.0018
	60	0.5	a: 0.5242; k_0_: 0.0125; b: 0.5196; k_1_: 0.0125	0.9900	0.0349	0.0015
		1.0	a: 0.5162; k_0_: 0.0045; b: 0.5154; k_1_: 0.0045	0.9889	0.0361	0.0015
		1.5	a: 0.2916; k_0_: 0.0041; b: 0.7427; k_1_: 0.0041	0.9875	0.0393	0.0018
	70	0.5	a: 0.0894; k_0_: 0.0143; b: 0.9624; k_1_: 0.0143	0.9915	0.0325	0.0016
		1.0	a: 0.6798; k_0_: 0.0071; b: 0.3719; k_1_: 0.0071	0.9885	0.0382	0.0017
		1.5	a: 0.1191; k_0_: 0.0056; b: 0.9296; k_1_: 0.0056	0.9909	0.0345	0.0014
Midilli	50	0.5	a: 0.9785; k: 0.0023; n: 1.2353; b: 0.0000	0.9976	0.0173	0.0004
		1.0	a: 0.9845; k: 0.0012; n: 1.1545; b: 0.0000	0.9975	0.0169	0.0003
		1.5	a: 0.9860; k: 0.0010; n: 1.1263; b: 0.0000	0.9979	0.0163	0.0003
	60	0.5	a: 0.9759; k: 0.0038; n: 1.2391; b: 0.0000	0.9985	0.0133	0.0002
		1.0	a: 0.9806; k: 0.0014; n: 1.1882; b: 0.0000	0.9980	0.0153	0.0003
		1.5	a: 0.9793; k: 0.0011; n: 1.2166; b: 0.0000	0.9972	0.0185	0.0004
	70	0.5	a: 0.9817; k: 0.0042; n: 1.2546; b: 0.0000	0.9990	0.0112	0.0002
		1.0	a: 0.9815; k: 0.0015; n: 1.2831; b: 0.0000	0.9985	0.0136	0.0002
		1.5	a: 0.9856; k: 0.0012; n: 1.2650; b: 0.0000	0.9988	0.0123	0.0002

**Table 3 foods-11-01751-t003:** Arrhenius-type equation and activation energy of the foams.

Foam Thickness (cm)	Arrhenius-Type Equation	R^2^	Activation Energy (kJ mol^−1^)	R^2^
0.5	$D_{\mathrm{eff}}=\left(1.2880{\times 10}^{-5}\right)\exp\left(-\frac{3032.5043}{\left(T+273.15\right)}\right)$	0.9291	25.2122	0.9282
1.0	$D_{\mathrm{eff}}=\left(4.0785\times 10^{-4}\right)\exp\left(-\frac{4017.0737}{\left(T+273.15\right)}\right)$	0.9840	33.3979	0.9742
1.5	$D_{\mathrm{eff}}=\left({2.355\times 10}^{-3}\right)\exp\left(-\frac{4403.3859}{\left(T+273.15\right)}\right)$	0.9855	36.6097	0.9829

**Table 4 foods-11-01751-t004:** Physicochemical properties of the foams dried under different drying conditions.

Drying Conditions	Color Parameters			Vitamin C (mg 100 g^−1^ Dry Basis)	Total Flavonoids (mg 100 g^−1^ Dry Basis)	Phenolic Compounds (mg 100 g^−1^ Dry Basis)
	L*	+a*	+b*			
Fresh Samples	41.55 ± 0.09 ^e^	15.51 ± 002 ^a^	3.87 ± 0.24 ^d^	34.15 ± 4.43 ^e^	122.64 ± 0.18 ^d^	2084.05 ± 6.51 ^f^
50 °C/0.5 cm	63.00 ± 0.62 ^a^	11.57 ± 0.11 ^e^	44.66 ± 0.79 ^c^	55.21 ± 3.05 ^c,d^	111.52 ± 0.00 ^g^	678.93 ± 8.59 ^d^
50 °C/1.0 cm	62.05 ± 0.13 ^a^	12.05 ± 0.11 ^d^	45.65 ± 1.14 ^c^	50.96 ± 1.86 ^d^	106.47 ± 0.45 ^i^	660.68 ± 8.99 ^d^
50 °C/1.5 cm	59.80 ± 0.81 ^b^	12.54 ± 0.19 ^c^	47.85 ± 0.26 ^a,b^	47.80 ± 2.00 ^d^	101.53 ± 0.31 ^j^	552.59 ± 7.48 ^e^
60 °C/0.5 cm	60.82 ± 0.27 ^b^	11.93 ± 0.05 ^d,e^	44.83 ± 0.19 ^c^	66.57 ± 1.74 ^c^	111.32 ± 0.00 ^h^	1072.64 ± 4.82 ^b^
60 °C/1.0 cm	59.90 ± 0.09 ^b^	12.61 ± 0.04 ^c^	46.13 ± 0.22 ^b,c^	66.70 ± 0.74 ^c^	117.98 ± 0.13 ^f^	985.18 ± 7.78 ^c^
60 °C/1.5 cm	57.16 ± 0.48 ^c^	12.90 ± 0.38 ^c^	48.59 ± 0.31 ^a^	68.21 ± 3.66 ^c^	118.48 ± 0.35 ^e^	691.44 ± 0.00 ^d^
70 °C/0.5 cm	60.16 ± 0.23 ^b^	12.08 ± 0.14 ^d^	45.86 ± 0.48 ^b,c^	76.08 ± 4.31 ^b^	150.25 ± 0.13 ^c^	1334.48 ± 9.00 ^a^
70 °C/1.0 cm	57.29 ± 0.15 ^c^	12.87 ± 0.08 ^c^	48.21 ± 1.44 ^a^	81.33 ± 4.12 ^b^	154.16 ± 0.35 ^b^	1302.12 ± 4.63 ^a^
70 °C/1.5 cm	55.55 ± 0.29 ^d^	13.51 ± 0.00 ^b^	49.26 ± 0.16 ^a^	82.38 ± 2.19 ^a^	161.51 ± 0.34 ^a^	1113.90 ± 5.73 ^b^

Values are means ± standard deviation of quadruplicate determination. Means with the same letter in the same column indicate no significant difference by Tukey test (*p* < 0.05).

## Data Availability

Data can be digitized from the graphs or requested to the corresponding author.

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
