# Peer review of "Mathematical Models to Describe the Foam Mat Drying Process of Cumbeba Pulp (Tacinga inamoena) and Product Quality"

_foods, 2022, doi:10.3390/foods11121751_

Round 1
Reviewer 1 Report
Following improvements needed:
1. There are some errors in the typing of equations, such as in equation (7) and (8); it needs to be corrected.
2. To show the accuracy of the mathematical model applied, the comparisons of the modelling results and the experimental data are strongly suggested to be visually shown. Hence, the experimental data are suggested to be presented in figure 1, followed by theoretical discussions.
3. The comparisons of the Deff calculated by Arrhenius type approximation and the values obtained based on experimental data are suggested to be presented in figure 3, then followed by theoretical discussions.
4. More comprehensive theoretical discussions/arguments, based on the mechanisms of the processes involved, are suggested to explain why the Midilli mathematical model results the best fit.
5. Theoretically, Deff depends more on the temperature of the foam and less on the temperature of the air, while the measured temperature is the temperature of the air. Since during drying the temperature of the material and the of the air are usually not equal, the use of the temperature of the air to be applied in the equations are suggested to be discussed.
6. The drying performance theoretically depends on the relative humidity of the air, so it is suggested also that the relative humidity of the air is to be reported.
Author Response
Response to Reviewer #1 Comments
Dear reviewer #1
We would like to thank you for your thoughtful comments, improving our manuscript.
Note: Besides the alterations explicitly mentioned, other modifications have been made (in red). The line numbers provided are referring to docx file with the new version of the article.
Following improvements needed:
- There are some errors in the typing of equations, such as in equation (7) and (8); it needs to be corrected.
- To show the accuracy of the mathematical model applied, the comparisons of the modelling results and the experimental data are strongly suggested to be visually shown. Hence, the experimental data are suggested to be presented in figure 1, followed by theoretical discussions.
- The comparisons of the Deff calculated by Arrhenius type approximation and the values obtained based on experimental data are suggested to be presented in figure 3, then followed by theoretical discussions.
Answer: Firstly, thanks for your positive comments (and initial scores). Equations (7) and (8) located between lines 126-131 have been corrected. We also tweaked Figure 1. We hope it looks better now.
Regarding “…the accuracy of the mathematical model applied, the comparisons…”, we added Figure 3 (located between lines 228-229) to the manuscript to show the accuracy of Midilli's mathematical model in describing the drying process of cumbeba pulp foam.
Regarding “...the comparisons of the Deff calculated by the Arrhenius-type approximation”, we present in Figure 4 (located between lines 258-259) the fit of the Arrhenius-type equation (Equation (7)) to the Deff data as a function of the drying temperature.
- More comprehensive theoretical discussions/arguments, based on the mechanisms of the processes involved, are suggested to explain why the Midilli mathematical model results the best fit.
Answer: Thank you for your comments. In the revised version of the article, we tried to demonstrate not only statistically, but also visually, that the Midilli model can be used accurately to represent the experimental data of the drying curves (please see lines 221-228) and hope the manuscript is better now.
- Theoretically, Deff depends more on the temperature of the foam and less on the temperature of the air, while the measured temperature is the temperature of the air. Since during drying the temperature of the material and the of the air are usually not equal, the use of the temperature of the air to be applied in the equations are suggested to be discussed.
Answer: We tried to improve the text and hope the manuscript is better now (see lines 240-245).
- The drying performance theoretically depends on the relative humidity of the air, so it is suggested also that the relative humidity of the air is to be reported
Answer: Thank you for your comments. Information on relative humidity has been added to the text (lines 88-89).
Sincerely,
Corresponding author
Reviewer 2 Report
In this paper authors have studied the foam drying kinetics and selected product quality of cumbeba pulp. This paper is well written in general and contains some important practical knowledge. However, I have a few comments.
On the chart authors should include the points with the standard deviation, not only lines connected to invisible points.
Lines 60-66. The specific information concerning used temperature and thickness should not be given here. It is repeated in the Materials and method section.
Line 90: Please give “-1+ in the superscript
How the equilibrium moisture content was determined?
Please check and correct the number of repetitions for individual experiments. For example, in the Materials and method section it is written that color was measured in six repetitions, whereas under the Table 4 that in four?
Author Response
Response to Reviewer 2 Comments
Dear reviewer #2
We would like to thank you for your thoughtful comments, improving our manuscript.
Note: Besides the alterations explicitly mentioned herein, other modifications have been made (in red). The line numbers provided are referring to docx file with the new version of the article.
In this paper authors have studied the foam drying kinetics and selected product quality of cumbeba pulp. This paper is well written in general and contains some important practical knowledge. However, I have a few comments.
On the chart authors should include the points with the standard deviation, not only lines connected to invisible points.
Answer: We thank reviewer #2 and agree with his opinion. We have included in Figure 1 (located between lines 181-182) the points with standard deviation.
Lines 60-66. The specific information concerning used temperature and thickness should not be given here. It is repeated in the Materials and method section.
Line 90: Please give “-1+ in the superscript.
Answer: Thanks for your comments. The text has been corrected (see lines 64-67 and line 88).
How the equilibrium moisture content was determined?
Answer: The text has been corrected and hope the manuscript is better now (lines 90-94).
Please check and correct the number of repetitions for individual experiments. For example, in the Materials and method section it is written that color was measured in six repetitions, whereas under the Table 4 that in four?
Answer: Thanks for your comments. The text has been corrected (see lines 140-141 and line 88)
Sincerely,
Corresponding author
Round 2
Reviewer 2 Report
The authors corrected the manuscript accordingly.